# Influence Analysis of Sustainability Perceptions on Sense of Community and Support for Sustainable Community Development in Relocated Communities

**DOI:** 10.3390/ijerph182212223

**Published:** 2021-11-21

**Authors:** Yi-Hsien Lin, Tsung-Hung Lee, Chiu-Kuang Wang

**Affiliations:** 1Department of Tourism, Ming Chuan University, Taoyuan 333023, Taiwan; yhlin@mail.mcu.edu.tw; 2Graduate School of Leisure and Exercise Studies, National Yunlin University of Science and Technology, Yunlin 640301, Taiwan; 3Department of Accounting, National Yunlin University of Science and Technology, Yunlin 640301, Taiwan; langalan0630@yahoo.com.tw

**Keywords:** sustainability perceptions, sense of community, sustainable community development, indigenous people, climate change

## Abstract

This study aims to examine a theoretical model using sustainability perceptions, including environmental; sociocultural; economic; and life satisfaction, sense of community, and support for sustainable community development among the indigenous people of two relocated communities in Taiwan. A total of 747 usable questionnaires were collected and analyzed using structural equation modeling. The analytical results indicated that sense of community is an antecedent of support for sustainable community development in both relocated communities. Life satisfaction perceptions can influence the sense of community in Rinari. Additionally, environmental and economic perceptions are antecedents of the sense of community in New Laiyi. Finally, this study provides theoretical implications to fill the gaps in previous research, and offers valuable insights for promoting residents’ support for sustainable community development in aboriginal communities; thus, this study has significant contributions, theoretically and practically.

## 1. Introduction

Since the mid-twentieth century, there have been numerous droughts, floods, and heat waves in many regions of the world, causing damage to human livelihoods and residential environments [1]. Over the past few decades, a mounting number of studies have indicated that extreme weather and climate events have become more intense and frequent [2,3].

A high frequency of extreme weather increases an individual’s level of concern over climate change [4], especially in regard to an adaptation to climate change for industry, land planning, and the livelihoods of residents [5]. Adaptation refers to actions taken by individuals, groups, and governments in different societies in response to past climate changes and in consideration of possible future climate threats. Adaptation can be motivated by the desire to protect economic well-being or to improve the level of safety [6]. Currently, climate variability influences governmental local natural resource management strategies to promote adaptive capacity and sustainable development [7].

In recent years, increasing research has focused on climate change adaptation in remote indigenous communities, examining issues such as improvements in the mental health of relocated residents [8]; the factors that influence residents’ relocation intentions [9]; residents’ perspectives on climate change and adaptation [10]; climate change vulnerability assessments [11]; and climate change adaptation planning [12]. However, discussions of relocated residents’ sustainability perceptions have been neglected, and the causal relationship between residents’ sustainability perceptions and support for sustainable community development remains unclear.

Relocated permanent housing and resettlement are common strategies employed for post-disaster development and reconstruction [13]. Previous studies have indicated that relocation has a negative impact on the society, culture, economy, and traditional concepts of indigenous communities [14]. Furthermore, relocation results in environmental, social, and psychological stress [15]. In environmental psychology, an individual’s perception reflects his or her evaluation or understanding of the environment based on personal experiences, beliefs, or cultural attitudes [16]. Scholars have suggested that an individual’s perception is a key element in supporting sustainable development [17,18]. However, no assessment of residents’ sustainability perceptions has yet been conducted, and the relationship between perceptions of sustainability and support for the sustainable development of a relocated community in an indigenous resettlement area is still undetermined.

The concept of sense of community is derived from the concept of sense of place [19]. Sense of community can be defined as the identification or emotional bond with certain people, groups, or areas by individuals who live in the same geographically defined areas, or by groups of people identified by common interests, values, and culture [20]. Past studies have suggested that sense of community is an important antecedent to an individual’s behavioral intentions in different research contexts [17,21]. Unfortunately, knowledge of the structural relationships among residents’ sustainability perceptions, sense of community, and support for sustainable community development in relocated indigenous communities is lacking.

This study aims to assess the structural relationships among perceptions and sustainable development in relocated indigenous communities using two different relocated indigenous settlements in southern Taiwan. In terms of theoretical and managerial contributions, this study proposes a theoretical model to fill the research gaps and provide managerial implications based on the analytical findings. In the sections that follow, we first present critical and recent studies on residents’ perceptions, sense of place, and community development, and then propose five hypotheses. Then, we apply the structural equation modeling (SEM) technique to build and examine the theoretical model. Finally, we discuss the theoretical and managerial implications of the model for the sustainable development of relocated indigenous communities. 

## 2. Theoretical Framework

### 2.1. Theoretical Background

Attachment theory is frequently applied to understand person–place relationships and the identity of certain places [22]. It regards an individual’s innermost emotional bond with particular individuals or places as a basic component of human nature [23]. Attachment theory is not only seen as a comprehensive theory of affective development, but is also used to explain an individual’s behavior in certain environments [24].

In attachment theory, an individual’s emotional bond and attitude are influenced by his or her experiences of certain places, such as his or her home, workplace, church, neighborhood, or city [24]. In this study, sense of community is considered a particularly exhaustive measure of the people–place relationship, including people’s social and emotional bonds with their community [25].

In the community research context, the concept of attachment theory has been widely applied to examine community residents’ emotional bonding with places as well as their behavioral intentions. For example, Lee [17] applied the concept of community attachment to examine residents’ support for sustainability, and the results suggested that community attachment was the critical factor affecting the level of support for sustainable tourism development. Stedman et al. [26] compared the residents of ten temperate-latitude lake district sites and elucidated the differences in attachment to the lakes at each site. The results indicated that every study of place attachment is a study of a single place and should remain tied to the site-specific environment, history, institutional structure, and culture. Scannell and Gifford [27] investigated the psychological benefits of place attachment and revealed thirteen categories of benefits derived from place attachment.

Previous studies have indicated that climate change might influence indigenous people’s livelihoods and perspectives on climate change and adaptation [10]. Based on attachment theory, residents’ emotions create a bond to a place based on their previous experiences of certain places [24], which means that sudden events or disasters associated with a place can influence their sense of place. In addition, individuals’ perceptions reflect the evaluations of the environment based on personal experiences [8]. Scholars have also indicated that individuals’ perceptions are a crucial factor influencing sustainable development [17,18]. Therefore, attachment theory seems to be an appropriate theoretical foundation for integrating the concepts of perceptions, sense of community, and support for sustainable community development, and constructing the theoretical model.

### 2.2. Sustainability Perceptions

Sustainable perception refers to an individual’s judgment or evaluation of environmental issues, or of an event based on personal experiences and attitudes toward certain environmental conditions [28]. In different research contexts, several studies have applied the concept of perception to explain individuals’ risk perceptions [29], environmental perceptions [30], and sustainability perceptions [18].

Past studies have indicated that the perception of sustainability is a crucial element in promoting support for sustainable development [31] and future behavior [32]. Recently, an increasing number of studies have applied different constructs to examine individuals’ sustainability perceptions. Several scholars have considered environmental perceptions, sociocultural perceptions, economic perceptions, and life satisfaction perceptions as important components of residents’ sustainability perceptions [18,31,33]. Environmental perceptions refer to the awareness of individuals of their surrounding environment and are one of the major research topics in environmental management [34]. Sociocultural perceptions are grounded in individuals’ own social and cultural experiences of everyday life, which reflect their emotional bond with their religion, traditions, and customs [35]. Economic perceptions involve individuals’ evaluation of the economic benefits of local industry and business [33]. Life satisfaction perceptions concern residents’ well-being, including health care, residential security, and daily life satisfaction [18].

The residents of indigenous communities have strong social networks, a close relationship with their land, and considerable family and kinship responsibilities [36]. Previous studies have indicated that traditional ecological knowledge practices and cultural and social relationships should also be seen as critical components to the relocation and adaption of indigenous people’s [37]. Moreover, some studies have found that perceptions can influence an individual’s attachment [19] and behavior [38]. However, some studies have also indicated that an individual’s attachment can affect his or her perceptions [17]; thus, the linear structural relationship among residents’ sustainability perceptions, sense of community, and support for sustainable community development remains undetermined.

### 2.3. Sense of Community 

Sense of community refers to individuals’ feelings of belonging and the interaction between people and their environment based on daily experiences [39]. Several scholars have indicated that a sense of place or community is a key issue in person–place relationships, human behaviors, and psychological research [19]. Thus, sense of community is regarded as the key theoretical construct in the research context of community psychology [39].

Sense of community is an important topic in community research and applied social psychology [40]. Several scholars have applied community indices in different research disciplines, such as social behavior [41]; business [42]; adolescent education [43]; tourism [21]; and environmental, recreational, and festival management [44]. To date, assessments of the relationship between sense of community and support for sustainable community development in relocated indigenous communities have been limited; thus, further studies are warranted.

### 2.4. Support for Sustainable Community Development

In recent years, sustainable development has been a critical issue in different research areas, such as festival management [45], tourism development [17], national park governance [46], and climate change adaptation [47]. Past studies have indicated that when individuals experience greater benefits than costs, including economic, social, cultural, and environmental concerns [18], they are more willing to support sustainable development [17]. Moreover, numerous studies have indicated that people’s emotional and psychological attachment to a certain place can result in a positive attitude toward environmentally friendly behavior and support for sustainable development [48]. Therefore, understanding residents’ emotional attachment to relocated communities has become an important issue for the administrators or developers of relocated indigenous settlements [49].

Over the past few decades, relocation after natural disasters has been a critical issue in post-disaster reconstruction in developing countries [50]. Relocated residents not only want to rebuild their houses or facilities but also yearn to maintain their livelihood, community, environment, and social connections in the relocated communities in which they now live [51]. Moreover, several studies have indicated that financial and social support should be considered as important factors for relocated residents [15]. Therefore, well-planned relocation can produce positive long-term developmental outcomes [13].

### 2.5. Hypothesized Model

In the psychological and behavioral research context, sense of community has become increasingly popular in the literature as a way to explain human social engagement [52]. Previous studies have investigated individuals’ perceptions as antecedents to their personal behavior and attitude [29]. Moreover, some studies have identified causal relationships between different perceptions and attachment/sense of community. For example, Bonaiuto et al. [53] indicated that sociodemographic and residential variables can influence residents’ attachment, and length of residence in the place and economic perceptions are the most relevant residential and sociodemographic variables. Rollero and De Piccoli [19] studied 328 residents from a city in northern Italy, and the results indicated that the individuals’ cultural level and social relationships were associated with place attachment. Mesch and Manor [16] argued that the local attachment of residents can result from a positive environmental perception. Stedman et al. [26] indicated that attachment to lakes is based, in part, on the perceived quality of life and perceptions of social conflict.

Although previous studies have indicated that social, environmental, physical, and emotional features can affect an individual’s sense of community or attachment to a place of residence [27], several theoretical models have been established to examine the predictors of the place attachment of residents in different places [19]. However, no studies have constructed a theoretical model to examine how individuals’ perceptions influence the sense of community in relocated indigenous communities.

Current studies have found that an individual’s perceptions of the natural and communal environment can influence the support for sustainable development [54]. The development of an indigenous community is based on its cultural heritage, environmental conditions, and residents’ well-being [55]. Several studies have indicated that positive perceptions of their community can improve residents’ intentions or behavior with regard to health care, environmental protection, and cultural inheritance [56,57]. Moreover, some studies have indicated that different perception variables (e.g., economic, sociocultural, environmental, and life satisfaction perceptions) can influence place attachment or sense of place [16,19,26,53].

Climate change adaptation, with regard to the issues of remote indigenous communities, has been considered an important topic in the sustainable development research context; however, the influence of residents’ sustainability perceptions through sense of community on support for sustainable community development in relocated indigenous communities has been neglected. To fill this research gap, we propose the following hypotheses:

**Hypothesis** **1** **(H1).**
*Economic perceptions positively and significantly affect sense of community.*


**Hypothesis** **2** **(H2).**
*Sociocultural perceptions positively and significantly affect sense of community.*


**Hypothesis** **3** **(H3).**
*Environmental perceptions positively and significantly affect sense of community.*


**Hypothesis** **4** **(H4).**
*Life satisfaction perceptions positively and significantly affect sense of community.*


In recent years, numerous studies in different research disciplines, such as tourism development [31], protected area management [46], and community development [58], have addressed residents’ support for sustainable development. Several studies have also indicated that an individual’s emotion, attitude, and attachment can influence his or her support for sustainable development [31].

A strong sense of community can improve residents’ well-being and increase their feelings of safety and security, their participation and involvement in community affairs, and their sense of civic responsibility [44]. Previous studies have indicated that for indigenous people, emotional bonds with the community and an attachment to the land in which they live play important roles in constructing the identity and coherence of their communities [59]. However, a relocation from the original place of residence to a resettlement can influence the emotional and social relationships of residents with the environment and neighbors of the new community [28]. To address this issue, the following hypothesis is proposed:

**Hypothesis** **5** **(H5).**
*Sense of community positively and significantly affects the support for sustainable community development.*


Integrating all the hypotheses specified above, we propose the theoretical model illustrated in Figure 1.

## 3. Methodology

### 3.1. Study Sites

In 2009, Typhoon Morakot struck Taiwan. It dropped 3000 mm of rain, leading to devastating mudslides and the worst flooding in Taiwan in 50 years [60]. Ultimately, more than 700 people died, and direct property losses amounted to over 3 billion USD [61]. Several indigenous communities in southern Taiwan, such as the Kucapungane, Makazayazaya, Paridrayan, and Tjalja’avus indigenous communities, were destroyed or seriously damaged by landslides caused by Typhoon Morakot [62].

Most indigenous communities in Taiwan are located in remote and mountainous areas [63,64] (for the indigenous peoples distribution map, also see Lee’s and Jan’s [63] map), which face natural disasters more often than urban areas. After Typhoon Morakot, the Taiwanese government passed the Morakot Post-disaster Reconstruction Special Act (MPDR Special Act). Based on the MPDR Special Act, several relocated communities were established for those who were displaced, including Rinari, Changchi Baihe, Ulaljuc, New Laiyi, Kuskus, and Central Road. By December 2014, nearly 5000 indigenous people had been moved to 6 relocated communities in Pingtung County. Of the six relocated communities, Rinari and New Laiyi are the two largest in terms of residential scale [65]; thus, we selected them as our study areas.

Rinari is located in Masiljid village, Makazayazaya Township, Pingtung County (22°49′70″ N, 120°86′65″ E), near the Ailiao River. It has 483 houses and more than 1400 residents [66]. Rinari includes three different indigenous communities for two different tribes: the Kucapungane community (Rukai tribe), the Tavalan community (Paiwan tribe), and the Makazayazaya community (Paiwan tribe). These communities and tribes have different ethnic backgrounds, historical rivalries, and customs. Currently, Rinari is considered one of the most famous indigenous tourism destinations [67,68].

New Laiyi is located in Wanlong village, Xinpi Township, Pingtung County (22°11′51″ N, 120°57′58″ E), near the Linbian River. It has 307 houses and nearly 1000 residents [69]. New Laiyi consists of the Tjana’asiya, Calasiv, and Tjalja’avus communities. In contrast to Rinari, all three communities belong to the same administrative area and the Paiwan tribe.

### 3.2. Research Instrument

Based on the literature review, this study applied environmental perceptions, sociocultural perceptions, economic perceptions, life satisfaction perceptions, sense of community, and support for sustainable community development as latent variables. A pilot survey was conducted in August 2019 in Rinari and New Laiyi. In total, 100 usable questionnaires were collected. Item analyses were applied to assess the quality of the questionnaire. Moreover, three scholars specializing in sustainable development in indigenous communities were invited to assess the content validity of the questionnaire. Based on the item analysis and the comments made by the three scholars, one item was deleted, and the wording of six items was slightly modified to improve comprehensibility. The formal questionnaire is described below.

This study applied environmental perceptions, sociocultural perceptions, economic perceptions, and life satisfaction perceptions as latent variables. In accordance with Lee and Jan [18], environmental perceptions were measured by 3 items, sociocultural perceptions by 5 items, economic perceptions by 4 items, and life satisfaction perceptions by 12 items.

Sense of community was measured by 12 items borrowed from Chavis et al. [70]. Sense of community included needs fulfillment (three items), membership (three items), influence (three items), and emotional connection (three items).

Support for sustainable community development was measured by three items borrowed from Lee [17]. It included supporting the relocated community’s sustainable development initiatives, participating in the relocated community’s sustainable development-related plans, and participating in the promotion of the relocated community’s environmental education and conservation.

All items were scored on a seven-point Likert scale, where one represented “strongly disagree” and seven represented “strongly agree”. Demographic variables, including gender, marital status, age, educational level, occupation, residential community before relocation, and ethnic group, were also recorded.

### 3.3. Data Collection

The questionnaire survey was conducted from August 2019 to March 2020 with residents of Rinari and New Laiyi. Four research assistants who were trained in the sampling technique were recruited to conduct the questionnaire survey. Before the questionnaire survey, we first visited the leader of each community (e.g., the tribal chieftain, chief of the village, and medium of the community), explained the purpose and the perspective of the study, and requested ethical clearance for the research. Finally, we received permission to conduct the study in the two communities and obtained ethical approval from the community leaders. Four residents in each community were hired to help the research assistants search for and identify community residents, ensure that the survey did not interfere with residents’ regular lives, and conduct the questionnaire survey. All respondents provided consent, were informed that their anonymity would be protected, and that data storage and usage limitations would be observed before they completed the questionnaire. Overall, 451 and 296 usable questionnaires were obtained from Rinari and New Laiyi, respectively.

### 3.4. Quality of the Research Instrument

The 451 and 296 responses from Rinari and New Laiyi were sufficient for confirmatory factor analysis (CFA) and SEM [71]. The Cronbach’s α values of all latent variables in the samples from Rinari (environmental perceptions = 0.89, sociocultural perceptions = 0.93, economic perceptions = 0.93, life satisfaction perceptions = 0.97, sense of community = 0.93, and support for sustainable community development = 0.86) and New Laiyi (environmental perceptions = 0.96, sociocultural perceptions = 0.92, economic perceptions = 0.99, life satisfaction perceptions = 0.97, sense of community = 0.95, and support for sustainable community development = 0.96) were above the basic criterion of 0.7 [72], indicating that the survey data had acceptable reliability.

### 3.5. Data Analysis

This study applied a multigroup invariance method to evaluate the significant differences between Rinari and New Laiyi. The results of the multigroup analysis (Table 1) indicated that the responses of the participants from the two study communities had significant differences [73]. Therefore, this study considered the responses from the two study communities as different groups and analyzed them separately.

In accordance with Anderson and Gerbing [74], this study applied two-stage analysis to assess the theoretical framework. In the first stage, CFA was applied to assess the measurement mode in terms of reliability, validity, and several model fit indices. In the second stage, SEM was applied to validate the research hypotheses, and all the parameters were analyzed by the maximum likelihood method.

## 4. Results

### 4.1. Reliability, Validity, and Common Method Variance Test

This study applied several methods to determine the reliability and validity of the measurement model. The multivariate normality assumption was evaluated using Mardia’s index test [75]. In this study, Mardia’s indices were below *p* (*p* + 2) in both study areas; therefore, the normality assumption was supported.

To evaluate the measurement model, several model fit indices were assessed, as shown in Table 2. All measurement model fit indices exceeded the suggested criteria; therefore, the measurement model fit the data in both study areas well [71,76,77].

Table 3 lists the factor loadings, t values, average variance extracted (AVE) values, and composite reliability (CR) values of the measurement model. All of the factor loadings were above 0.5 and were significant (t > 1.96, *p* < 0.001), and all AVE values and CR values exceeded the suggested thresholds (0.5 and 0.7, respectively), indicating that the measurement model had internal consistency and adequate convergent validity [71].

To test the discriminant validity of the measurement model, Table 4 lists the correlations between each latent variable and the square root of the AVE values. The correlation coefficients between the latent variables in New Laiyi are all below the square root of the AVE. However, some of the correlation coefficients between the latent variables in Rinari are greater than the square root of the AVE. Therefore, this study applied the confidence interval method to reconfirm the discriminant validity of the measurement model. As shown in Table 5, none of the indices in the 95% confidence interval of the correlation coefficient include 1.0, indicating that the measurement model has adequate discriminant validity without multicollinearity problems [78].

### 4.2. Structural Model

This study applied SEM analyses to assess the causal relationships among economic perceptions, sociocultural perceptions, environmental perceptions, life satisfaction perceptions, sense of community, and support for sustainable community development. The model fit indices of the structural model for both study areas are presented in Table 2. All model fit indices indicate an acceptable fit for the structural model in both study areas [71].

Figure 2 and Figure 3 illustrate the structural model for the residents of Rinari and New Laiyi, respectively. For the residents of Rinari, life satisfaction perceptions positively and significantly affected sense of community, and sense of community positively and significantly affected the support for sustainable community development. Thus, H4 and H5 were supported. However, the effects of economic perceptions, sociocultural perceptions, and environmental perceptions on sense of community were not significant; therefore, H1, H2, and H3 were rejected.

For the residents of New Laiyi, economic perceptions and environmental perceptions positively and significantly affected sense of community, and sense of community positively and significantly affected the support for sustainable community development. Thus, H1, H3, and H5 were supported. The effects of sociocultural perceptions and life satisfaction perceptions on sense of community were negatively significant; thus, H2 and H4 were rejected.

## 5. Discussion

### 5.1. Theoretical Implications

The formation of residents’ sense of community or place attachment has been a critical research topic in community management and place development [40,48]. Indigenous people have stronger connections and emotional bonds with their place of residence and community than non-indigenous people [79]. In this study, sense of community is found to be an antecedent of support for sustainable community development in both relocated communities, which is similar to the results of previous studies [17]. In the place and community research context, sense of community, sense of place, and place attachment are important factors influencing residents’ activities and behaviors. However, only a few studies have assessed the place attachment or sense of community of residents of indigenous communities [60,80], and no study has examined the causal relationships between sense of community and different variables in a relocated indigenous community. Therefore, this study not only finds that sense of community plays an important role in constructing an individual’s support for sustainable community development in a relocated indigenous community, but also provides a valuable reference for relocated community development and climate change adaptation among indigenous people, potentially contributing to the literature.

In Rinari, life satisfaction perceptions were an antecedent to sense of community, which is a finding similar to that in previous studies [26]. However, economic perceptions, sociocultural perceptions, and environmental perceptions did not directly influence the sense of community of residents in Rinari. According to attachment theory [22], the attachment of individuals is influenced by their feelings about, bond with, and childhood experiences of a certain place. Accordingly, the analytical results for Rinari may have been due to its sufficient infrastructures, beautiful views, good reputation, and community school. When residents have more place satisfaction and attachment, they are more willing to participate in place affairs and sustainable community development [81]. Therefore, the results for Rinari obtained by this study support the view that place satisfaction can influence residents’ attachment, which, in turn, enhances their sustainable behavior.

In New Laiyi, economic perceptions and environmental perceptions were antecedents to sense of community, which is in agreement with the results of several previous studies [53,80]. However, sociocultural perceptions and life satisfaction perceptions did not directly influence residents’ sense of community in New Laiyi. These analytical results might be explained by the location of the resettlement site. New Laiyi is located near the Chaozhou township, the second largest township in Pingtung County, with more than 53000 residents [82]. A large city can provide more working opportunities, which, in turn, improve the financial status and spending power of families. Several facilities that were considered iconic features in the indigenous community, such as farms, flagstone houses, thatched granaries, and taro kilns, were reconstructed in New Laiyi [83]. These facilities may have not only maintained the community’s traditional customs, but also improved residents’ connection with the place [84,85].

In this study, sociocultural perceptions did not affect the sense of community of residents in the two relocated communities. This finding is different from those of previous studies [19]. Indigenous cultures have an inseparable tie to traditional tribal territory [86], including ritual land, ancestral holy land, and community, hunting and farming areas [87]. In an indigenous community, land is an important component for establishing an attachment to the community and for developing traditional culture [88]. When indigenous people relocate from the original community to a resettlement area, they are disconnected from their traditional lands and territory. Therefore, in this early stage, they cannot establish their sense of community through sociocultural perceptions.

There were several differences in the analytical results between the two communities. New Laiyi is located near the Chaozhou township, which provides more working opportunities; however, without the community’s economic activities and community school, life satisfaction perceptions cannot be improved. In Rinari, the tourism industry is more developed because of the excellent surrounding views and featured buildings. Indigenous handicrafts drive the development of the cultural and creative industry, and tourists can experience indigenous cuisine. However, not all residents are involved in the tourism industry or share in the benefits from it. Thus, economic perceptions cannot be established in Rinari.

According to attachment theory, different life experiences can influence individuals’ sense of community and influence their behavioral intention [24]. In this study, we found that different locations, psychological and physical benefits, and established schools in the two indigenous communities can influence the daily life experiences and perceptions of residents, leading to different effects on the sense of community in the two different study sites. This study thus fills research gaps by extending the application of attachment theory in the community research context.

### 5.2. Managerial Implications

The analytical findings indicate that in the two study areas, sense of community is an antecedent of support for sustainable community development. Sense of community reflects the feeling of belonging and emotional bonds with a certain place and is seen as an important component of the person–land relationship [39]. Therefore, this study suggests that administrators or managers in both communities should consider enhancing the psychological benefits provided by the communities, constructing more public open spaces, and improving neighborhood ties [27,44]. For example, administrators can establish more community care centers for elderly individuals or playgrounds for children. Additionally, they can provide more courses, such as flower arrangement, cooking, and athletic courses, which not only enhance psychological benefits but also provide more opportunities to interact with neighbors. Furthermore, administrators can consider establishing police stations to improve community security [27], which can also enhance the residents’ sense of community.

This study suggests that administrators or developers can consider establishing shuttle bus routes to connect the communities to railway stations or cities (e.g., the Chaozhou Township or Pintung City). Shuttle buses not only provide convenient transportation to large cities, but also extend the opportunities for tourists to visit Rinari and enhance tourism industry development [89]. Moreover, administrators or developers should provide more environmental education courses for residents, as such courses can improve residents’ knowledge of the environment of their community, which, in turn, can generate positive environmental perceptions [90]. Additionally, Rinari contains three different communities within two different tribes, and each community or tribe has its own customs and cultural background. Thus, we suggest that in Rinari, each community and tribe should develop its own ritual celebrations and its own customs and culture, as such developments can improve residents’ cultural identity and the preservation of the community.

Our suggestions for the administrators or developers of New Laiyi are as follows. First, as shown in previous research, schools play an important role in improving residents’ interpersonal relationships and coherence with the community [91]. Community schools can serve as public open spaces and provide residents with more activities for leisure or social interaction, which, in turn, can enhance their life satisfaction perceptions [92]. It is thus suggested that administrators or developers should establish a community school in New Laiyi. Second, administrators or developers should organize more cultural activities for residents, such as courses on sculpture, hunting, tribal language, or ritual celebrations, as such courses can enhance the cultural knowledge of residents [93].

### 5.3. Limitations and Future Research Directions

Although our findings potentially make several contributions, there are several limitations that should be addressed in future research. First, this study applied four variables to measure the sustainability perceptions of residents. However, several studies have proposed other components of individuals’ sustainability perceptions. For example, Gattig and Hendrickx [94] argued that the residents’ risk perceptions, including environmental, financial, and health risks, are important components of sustainability perceptions. Holladay and Powell [95] proposed social–ecological resilience, including trust, networks, local control, flexible governance, leakage prevention, and controlled infrastructure development, to assess sustainability perceptions. Further research using these variables to more precisely assess the sustainability perceptions of residents is recommended.

Second, although this study provides several theoretical and managerial implications for two relocated communities, these suggestions might not be applicable to other relocated communities in Taiwan and other countries. To overcome this limitation, future research should assess the same theoretical model in relocated communities in different countries to verify its generalizability and to provide multicultural and/or international perspectives [96].

Finally, this study applied a self-reported survey to test a theoretical model. However, individuals’ perceptions can be influenced by their personal experiences [28], and some psychological or emotional attitudes toward community development may not actually be reflected in quantitative research methods [18]. To overcome this shortcoming, we recommend that future research uses qualitative methods, such as in-depth interviews, focus groups, and participatory observations, to better capture the details of sustainable community development and the personal experiences that influence residents’ sustainability perceptions [18].

## 6. Conclusions

Although climate change adaptation in remote indigenous communities has become a critical issue in the context of research on indigenous people, there has been no assessment of a structural model of residents’ sustainability perceptions, sense of community, and support for sustainable community development among residents in relocated indigenous communities. This study develops a theoretical model of economic perceptions, sociocultural perceptions, environmental perceptions, life satisfaction perceptions, sense of community, and support for sustainable community development among resettled indigenous people in Taiwan. The analytical results indicate that life satisfaction perceptions have a direct effect on the sense of community in Rinari and that economic perceptions and environmental perceptions have a direct effect on the sense of community in New Laiyi. Moreover, sense of community has a direct effect on support for sustainable community development in both relocated communities. Therefore, this study suggests that administrators and developers should consider the different demands of each relocated community to improve the positive sustainability perceptions of residents, which, in turn, will increase their sense of community.

This study potentially makes contributions with both theoretical and practical implications and expands our knowledge in the research context of sustainability perceptions and the applicability of attachment theory in the community research context. The original theoretical framework proposed in this study sheds light on behavioral models for promoting support for sustainable community development in relocated indigenous communities.

## Figures and Tables

**Figure 1 ijerph-18-12223-f001:**
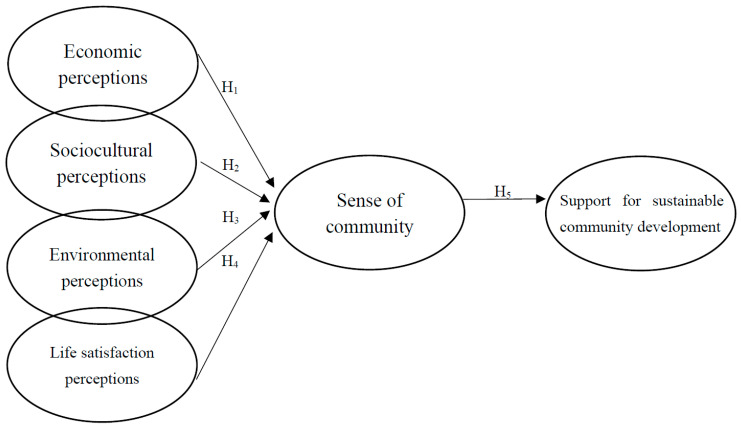
Theoretical model.

**Figure 2 ijerph-18-12223-f002:**
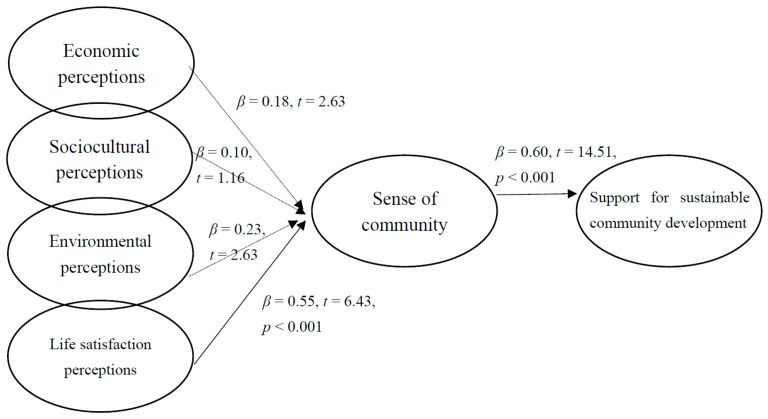
Structural model for Rinari.

**Figure 3 ijerph-18-12223-f003:**
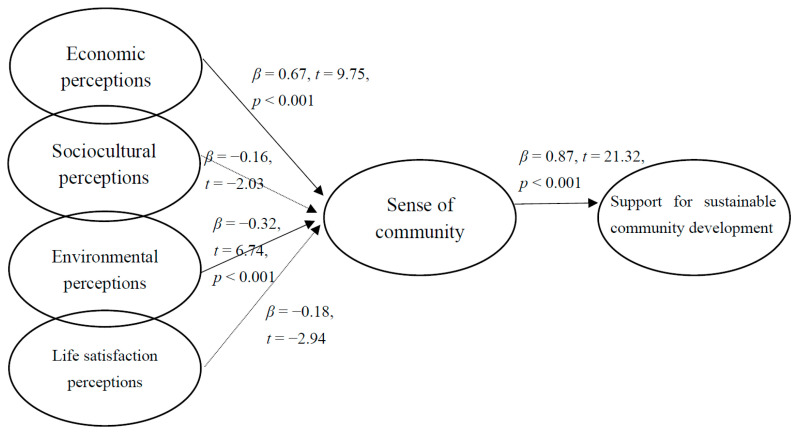
Structural model for New Laiyi.

**Table 1 ijerph-18-12223-t001:** Multigroup invariance.

Model	df	Δdf	χ^2^	Δχ^2^	*p*
Unconstrained	1372		6096.144		0.000
Measurement weights	1405	33	6184.145	88.001	0.000
Measurement intercepts	1444	72	7330.642	1234.497	0.000
Structural covariances	1465	93	7797.39	1701.246	0.000
Measurement residuals	1505	133	10,839.011	4742.867	0.000

**Table 2 ijerph-18-12223-t002:** Goodness of fit.

Indices	Criteria	Measurement Model Fit		Structural Model Fit		References
		Rinari	New Laiyi	Rinari	New Laiyi	
χ^2^/df	<3	2.55	1.825	2.326	1.733	Hair et al. (2010)
GFI	>0.80	0.871	0.881	0.884	0.884	McDonald and Ho (2002)
AGFI	>0.80	0.84	0.828	0.856	0.837	McDonald and Ho (2002)
RMSEA	<0.08	0.059	0.053	0.054	0.05	McDonald and Ho (2002)
SRMR	<0.08	0.047	0.63	0.044	0.076	Hu and Bentler (1999)
NFI	>0.90	0.932	0.962	0.938	0.963	Hu and Bentler (1999)
CFI	>0.95	0.957	0.982	0.964	0.984	Hu and Bentler (1999)

GFI = Goodness of Fit Index, AGFI = Adjusted Goodness of Fit Index, RMSEA = Root Mean Square Error of Approximation, SRMR = Standardized Root Mean Square Residual, NFI = Normed-Fit Index, CFI = Comparative Fit Index.

**Table 3 ijerph-18-12223-t003:** Outcomes of confirmatory factor analysis.

	Factor Loading		T -Value		AVE		CR	
	Rinari	New Laiyi	Rinari	NewLaiyi	Rinari	New Laiyi	Rinari	NewLaiyi
**Economic perceptions**					0.76	0.96	0.93	0.99
Increase employment opportunities	0.89	0.99	23.97	24.20				
Increase shopping opportunities	0.94	0.99	26.44	24.50				
Increase local government tax revenues	0.78	0.97	19.45	23.51				
Promote local business opportunities	0.86	0.96	22.77	22.86				
**Socio-cultural perceptions**					0.74	0.85	0.93	0.97
Participate in cultural activities	0.85	0.92	22.25	21.36				
Develop cultural activities	0.86	0.95	22.74	22.80				
Preserve the local culture	0.84	0.96	21.95	23.29				
Cultural exchanges	0.88	0.93	23.39	21.85				
Positive effects on cultural identity	0.85	0.84	22.45	18.44				
**Environmental perceptions**					0.73	0.90	0.89	0.96
Protect the natural environment	0.86	0.95	22.46	22.42				
Protect the human/cultural resources	0.85	0.97	21.84	23.46				
Increase environmental awareness	0.85	0.92	22.16	21.57				
**Life satisfaction perceptions**					0.72	0.78	0.97	0.98
Health well-being	0.88	0.90	23.68	22.72				
Safety well-being	0.86	0.93	22.64	23.62				
Family satisfaction	0.85	0.94	22.34	23.98				
Satisfaction with leisure	0.79	0.92	19.91	22.85				
Satisfaction with spiritual life	0.85	0.95	22.11	24.67				
Satisfaction with cultural life	0.85	0.96	22.31	24.90				
Satisfaction with social life	0.87	0.94	23.38	24.54				
Satisfaction with neighbors	0.84	0.84	21.70	19.67				
Satisfaction with housing	0.85	0.68	22.09	14.81				
Standard of living	0.86	0.79	22.72	12.61				
Life is excellent	0.85	0.87	22.09	21.38				
Overall life satisfaction	0.81	0.86	20.90	20.88				
**Sense of community**					0.54	0.70	0.93	0.96
This relocated community is a good place to live	0.72	0.59	16.87	9.79				
Neighbors and I have the same ideas about the relocated community	0.76	0.73	18.52	14.76				
People in this relocated community share the same values	0.63	0.79	14.61	16.85				
I belong to this relocated community	0.79	0.96	19.55	22.05				
I can recognize most of the people who live in the relocated community	0.74	0.86	17.81	19.16				
Most of my neighbors know me	0.71	0.84	16.88	18.34				
I have a say about what goes on in the relocated community	0.78	0.88	19.23	19.75				
If there is a problem in this relocated community, people who live here can get it solved	0.74	0.87	18.21	19.15				
I care about what my neighbors think of my actions	0.72	0.83	17.31	17.94				
I have a good bond with others in this relocated community	0.83	0.92	20.90	21.22				
It is very important to me to live in this relocated community	0.75	0.93	18.47	21.57				
I expect to live in this relocated community for a long time	0.62	0.75	14.21	15.12				
**Support for community sustainable development**					0.70	0.90	0.87	0.96
Support the relocated community sustainable development initiatives	0.88	0.97	23.31	23.25				
Participate in relocated community sustainable plans and development	0.94	0.99	25.66	24.65				
Participate in the promotion of environmental education and conservation	0.67	0.88	15.62	19.85				
Mardia’s coefficient	514.15	300.88						
*p* (*p* + 2)	1599.00							

*p* = number of items.

**Table 4 ijerph-18-12223-t004:** Correlation coefficients and AVE matrix.

	ECPs		SCPs		ENPs		LSPs		SOC		SSCD	
	Rinari	New Laiyi	Rinari	New Laiyi	Rinari	New Laiyi	Rinari	New Laiyi	Rinari	New Laiyi	Rinari	New Laiyi
ECPs	0.871	0.979										
SCPs	0.809	0.790	0.858	0.921								
ENPs	0.750	0.517	0.879	0.474	0.854	0.949						
LSPs	0.699	0.440	0.859	0.637	0.889	0.476	0.846	0.885				
SOC	0.643	0.601	0.705	0.347	0.735	0.525	0.748	0.166	0.734	0.832		
SSCD	0.460	0.615	0.518	0.517	0.512	0.606	0.601	0.475	0.640	0.769	0.837	0.948

ECPs = Economic Perceptions; SCPs = Sociocultural Perceptions; ENPs = Environmental Perceptions; LSPs = Life Satisfaction Perceptions; SOC = Sense of Community; and SSCD = Support for Sustainable Community Development. Square root of AVE is shown on the diagonal of the matrix.

**Table 5 ijerph-18-12223-t005:** Discriminant validity of confidence interval.

Parameters	Point Estimate		SE		Point Estimate ± 2SE				Bias-Corrected				Percentile Method			
					Lower		Upper		Lower		Upper		Lower		Upper	
	Rinari	New Laiyi	Rinari	New Laiyi	Rinari	New Laiyi	Rinari	New Laiyi	Rinari	New Laiyi	Rinari	New Laiyi	Rinari	New Laiyi	Rinari	New Laiyi
ECPs–SCPs	0.81	0.80	0.03	0.02	0.76	0.76	0.86	0.85	0.76	0.75	0.86	0.84	0.76	0.75	0.86	0.84
ECPs–ENPs	0.75	0.47	0.03	0.04	0.68	0.39	0.81	0.56	0.68	0.39	0.80	0.55	0.68	0.39	0.80	0.55
ECPs–LSPs	0.69	0.58	0.03	0.04	0.63	0.50	0.75	0.66	0.63	0.50	0.74	0.65	0.63	0.50	0.75	0.65
ECPs–SSCD	0.43	0.68	0.05	0.03	0.33	0.62	0.53	0.74	0.32	0.62	0.52	0.73	0.32	0.62	0.52	0.73
ECPs–SOC	0.62	0.56	0.04	0.04	0.54	0.48	0.69	0.64	0.54	0.48	0.69	0.64	0.54	0.48	0.69	0.64
SCPs–ENPs	0.90	0.39	0.02	0.06	0.86	0.28	0.94	0.51	0.85	0.28	0.93	0.50	0.86	0.27	0.93	0.50
SCPs–LSPs	0.86	0.76	0.02	0.03	0.83	0.70	0.90	0.82	0.83	0.70	0.89	0.82	0.83	0.69	0.89	0.81
SCPs–SOC	0.67	0.29	0.04	0.06	0.59	0.17	0.75	0.42	0.58	0.17	0.74	0.42	0.58	0.16	0.74	0.42
SCPs–SSCD	0.51	0.55	0.05	0.04	0.41	0.46	0.61	0.64	0.41	0.46	0.60	0.63	0.41	0.46	0.60	0.63
ENPs–LSPs	0.88	0.42	0.02	0.06	0.85	0.30	0.91	0.55	0.85	0.29	0.91	0.53	0.85	0.29	0.91	0.54
ENPs–SOC	0.70	0.50	0.04	0.06	0.62	0.39	0.78	0.62	0.62	0.38	0.77	0.61	0.62	0.39	0.77	0.61
ENPs–SSCD	0.50	0.55	0.05	0.04	0.41	0.46	0.60	0.63	0.41	0.46	0.60	0.63	0.41	0.46	0.60	0.62
LSPs–SOC	0.68	0.11	0.04	0.07	0.60	−0.03	0.75	0.25	0.60	−0.02	0.75	0.25	0.60	−0.03	0.75	0.24
LSPs–SSCD	0.59	0.48	0.04	0.05	0.50	0.38	0.68	0.58	0.51	0.38	0.68	0.58	0.50	0.38	0.68	0.58
SOC–SSCD	0.59	0.78	0.05	0.03	0.50	0.72	0.68	0.84	0.50	0.71	0.68	0.84	0.50	0.71	0.67	0.84

ECPs = Economic Perceptions; SCPs = Sociocultural Perceptions; ENPs = Environmental Perceptions; LSPs = Life Satisfaction Perceptions; SOC = Sense of Community; and SSCD = Support for Sustainable Community Development.

## Data Availability

The original contributions presented in the study are included in the article; further inquiries can be directed to the corresponding author.

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
