# Peer review of "Influence Analysis of Sustainability Perceptions on Sense of Community and Support for Sustainable Community Development in Relocated Communities"

_ijerph, 2021, doi:10.3390/ijerph182212223_

Round 1

Reviewer 1 Report

I feel that some of the richer material about the relocated communities and the discussion/findings are somewhat lost in the quantitative analysis and description of the methodology. I would suggest some improvements as follows:

The abstract could be written better with less repetition of the word 'perception' and acronyms - use these in the main text. The final sentence of the abstract is rather general and perhaps overclaiming what the research has found out....

The introduction reads like descriptive, discrete paragraphs rather than leading the reader into what is to come. There needs to be a connection between the paragraphs and state clearly the aims of the research and the research questions that you are addressing and why the research is needed/what is unique about it. Then tell the reader what is coming next in subsequent sections. 

The use of so many acronyms is confusing and it is not clear what RSP means and it is mentioned early on in the main text. I would revisit the use of so many acronyms - can you reduce the amount and write in plain text instead? 

The paper does not add to theoretical understandings as there is not a theoretical framework. What theorists do you draw on to understand for example community, sustainability, sustainable communities, sustainable development and how do these all relate to climate change, risks and disasters and the relocation of communities. You also refer to aboriginal communities in Taiwan for which there must be a rich literature - you assume the reader know about this? A map would be useful. Also, think about terminology - is aboriginal used or indigenous people or something else? 

This leads to the ethical considerations of the research which are not mentioned at all. There needs to be a section on the ethical processes and considerations. Did you have to get ethical clearance for the research? issues of gaining access to the communities, informed consent, anonimity, data storage etc. Any ethical concerns/issued raised by the participants and how did you address these?  

Overall, I suggest thinking about what the paper is about and what you can claim to have found out. Then tell the reader clearly - a lot of numbers do not tell me anything.

Author Response

Reviewer 1

Overall Comments: I feel that some of the richer material about the relocated communities and the discussion/findings are somewhat lost in the quantitative analysis and description of the methodology. I would suggest some improvements as follows:

Responses from Authors: Thank you for offering this opportunity to improve this study (ijerph-1406919) from your helpful comments. We have addressed each comment in details, and highlighted the changes in the revised manuscript using blue colored text. Our responses to your comments are as follows:

  1. Thank you for your helpful comment. We have revised whole manuscript to reduce the word 'perception' and remove all acronyms. In addition, we have revised the abstract to avoid overclaiming what the research has found out. We revised the abstract as follows:

(P1; line 10-20)

This study aims to examine a theoretical model using sustainability perceptions including environmental, sociocultural, economic and life satisfaction; sense of community; and support for sustainable community development among the Indigenous people of two relocated communities in Taiwan. A total of 747 usable questionnaires were collected and analyzed using structural equation modeling. The analytical results indicated that sense of community is an antecedent of support for sustainable community development in both relocated communities. Life satisfaction perceptions can influence the sense of community in Rinari. Additionally, environmental and economic perceptions are antecedents of the sense of community in New Laiyi. Finally, this study provides theoretical implications to fill the gaps in previous research, and offers valuable insights for promoting residents’ support for sustainable community development in aboriginal communities, thus, this study has significant contributions theoretically and practically.

  1. According to your suggestion, we have revised the introduction to improve the readability and clarify the aims of the research. We have also added the paragraph to tell the reader what is coming next in subsequent sections.

(PP. 1-2; line 26-30, 39-46, 54-57, 67-76)

Since the mid-twentieth century, there have been numerous droughts, floods, and heat waves in many regions of the world, causing damage to human livelihoods and residential environments [1]. Over the past few decades, a mounting number of studies have indicated that extreme weather and climate events have become more intense and frequent [2, 3].

In recent years, increasing research has focused on climate change adaptation in remote Indigenous communities, examining issues such as improvements in the mental health of relocated residents [8], the factors that influence residents’ relocation intentions [9], residents’ perspectives on climate change and adaptation [10], climate change vulnerability assessments [11] and climate change adaptation planning [12]. However, discussions of relocated residents’ sustainability perceptions have been neglected, and the causal relationship between residents’ sustainability perceptions and support for sustainable community development remains unclear.

However, no assessment of residents’ sustainability perceptions has yet been conducted, and the relationship between perceptions of sustainability and support for the sustainable development of a relocated community in an Indigenous resettlement area is still undetermined.

This study aims to assess the structural relationships among perceptions and sustainable development in relocated Indigenous communities using two different relocated Indigenous settlements in southern Taiwan. In terms of theoretical and managerial contributions, this study proposes a theoretical model to fill the research gaps and provide managerial implications based on the analytical findings. In the sections that follow, we first present critical and recent studies on residents’ perceptions, sense of place, and community development and then propose five hypotheses. Then, we apply the structural equation modeling (SEM) technique to build and examine the theoretical model. Finally, we discuss the theoretical and managerial implications of the model for relocated Indigenous communities’ sustainable development.

  1. According to your suggestion, we have removed all acronyms in main text and write in plain text.

  1. According to your suggestion, we have replenished the theoretical background in section 2.1. In this study, we applied attachment theory as theoretical framework, and elucidated that the attachment theory can be regarded an appropriate theoretical foundation to integrate the concepts of perceptions, sense of community, and support for sustainable community development to construct the theoretical model. The revised manuscript sees as follow:

(PP. 2-3, line 78-109)

2.1 Theoretical Background

Attachment theory is frequently applied to understand person-place relationships and the identity of certain places [22]. It regards an individual’s innermost emotional bond with particular individuals or places as a basic component of human nature [23]. Attachment theory is not only seen as a comprehensive theory of affective development but is also used to explain an individual’s behavior in certain environments [24].

In attachment theory, an individual’s emotional bond and attitude are influenced by his or her experiences of certain places, such as his or her home, workplace, church, neighborhood or city [24]. In this study, sense of community is considered a particularly exhaustive measure of the people-place relationship, including people’s social and emotional bonds with their community [25].

In the community research context, the concept of attachment theory has been widely applied to examine community residents’ emotional bonding with places as well as their behavioral intentions. For example, Lee [17] applied the concept of community attachment to examine residents’ support for sustainability, and the results suggested that community attachment was the critical factor affecting the level of support for sustainable tourism development. Stedman et al. [26] compared the residents of ten temperate-latitude lake district sites and elucidated the differences in attachment to the lakes at each site. The results indicated that every study of place attachment is a study of a single place and should remain tied to the site-specific environment, history, institutional structure and culture. Scannell and Gifford [27] investigated the psychological benefits of place attachment and revealed thirteen categories of benefits derived from place attachment.

Previous studies have indicated that climate change might influence Indigenous people’s livelihoods and perspectives on climate change and adaptation (10). Based on attachment theory, residents’ emotions create a bond to a place based on their previous experience of certain places [24], which means that sudden events or disasters associated with a place might influence their sense of place. In addition, individuals’ perceptions reflect evaluations of the environment based on personal experiences [8]. Scholars have also indicated that individuals’ perceptions are a crucial factor influencing sustainable development [17, 18]. Therefore, attachment theory seems to be an appropriate theoretical foundation for integrating the concepts of perceptions, sense of community, and support for sustainable community development and constructing the theoretical model.

  1. According to your suggestion, we have elucidated the location characters of Indigenous community distribution, and suggest readers to refer to Lee and Jan’s (2021) Indigenous people distribution map. In addition, we have used the term “Indigenous” to replace aboriginal in the manuscript. We revised the manuscript as follows:

(P 6; line 258-260)

Most Indigenous communities in Taiwan are located in remote and mountainous areas [64, 65] (for the Indigenous peoples distribution map, also see Lee and Jan’s [64] figure 2), which face natural disasters more often than urban areas.

  1. Thank you for your helpful comment. Before the questionnaire survey conduct, we have visited the leader of two communities to make ethical clearance for the research, and received the permission to conduct the study. In addition, four residents in each community were hired to ensure the survey did not bother residents’ regular life and conduct the questionnaire survey. Each respondent was asked the consent before filled out the questionnaire. We have replenished 3.3. Data Collection as follows:

(P. 7; line 311-320)

Before the questionnaire survey, we first visited the leader of each community (e.g., the tribal chieftain, chief of the village, and medium of the community), explained the purpose and the perspective of the study, and requested ethical clearance for the research. Finally, we received permission to conduct the study in the two communities and obtained ethical approval from the community leaders. Four residents in each community were hired to help the research assistants search for and identify community residents, ensure that the survey did not interfere with residents’ regular lives and conduct the questionnaire survey. All respondents provided consent, and they were informed that their anonymity would be protected and that data storage and usage limitations would be observed before they completed the questionnaire.

  1. Thank you for your helpful comment. The discussion and conclusions section have been revised. We have added more convincing arguments regarding the contributions of the manuscript, and discuss the different results obtained for the two analyzed sites.

(PP. 12-14; line 472-483, 502-508, 582-591)

In New Laiyi, economic perceptions and environmental perceptions were antecedents of sense of community, which is in agreement with the results of several previous studies [53, 81]. However, sociocultural perceptions and life satisfaction perceptions did not directly influence residents’ sense of community in New Laiyi. These analytical results might be explained by the location of the resettlement site. New Laiyi is located near Chaozhou township, the second-largest township in Pingtung County, with more than 53 thousand residents [83]. A large city can provide more working opportunities, which in turn improve the financial status and spending power of families. Several facilities that were considered iconic features in the Indigenous community, such as farms, flagstone houses, thatched granaries, and taro kilns, were reconstructed in New Laiyi [84]. These facilities may have not only maintained the community’s traditional customs but also improved residents’ connection with the place [85, 86].

According to attachment theory, different life experiences might influence individuals’ sense of community and influence their behavioral intention [24]. In this study, we found that different locations, psychological and physical benefits, and established schools in the two Indigenous communities might influence residents’ daily life experiences and perceptions, leading to different effects on the sense of community in the two different study sites. This study thus fills research gaps by extending the application of attachment theory in the community research context.

Therefore, this study suggests that administrators and developers should consider the different demands of each relocated community to improve the positive sustainability perceptions of residents, which in turn will increase their sense of community.

This study potentially makes contributions with both theoretical and practical implications and expands our knowledge in the research context of sustainability perceptions and the applicability of attachment theory in the community research context. The original theoretical framework proposed in this study sheds light on behavioral models for promoting support for sustainable community development in relocated Indigenous communities.

Reviewer 2 Report

The study is about to investigate the influence of sustainability perceptions on sense of community and support for sustainable community development in two Taiwanese relocated communities.

The study proved that sustainability perceptions ifluencing the sense of community which can be considered as an antecedent of SSCD in both relocated communities. The authors found differences between the studied communities in terms of components that have a direct effects on the SOC.

The topic of the paper is relevant and important. It is a fundamental issue for communities that are exposed to adverse weather conditions today and even more so in the future.

Originality

Research contribution in the paper can be identified, so, the results provide an advance in current knowledge. The study provides new insights and new knowledge on the topic in question.

Title

The title looks good at first glance but something is missing. I suggest changing the title by adding "in relocated communities". In this case, the title would be: “Influence Analysis of Sustainability Perceptions on Sense of Community and Support for Sustainable Community Development in relocated communities”, which expresses the orientation of the article more and is more in line with the content.

Anyhow, the title is clear and it is adequate to the content of the article. The study, also in terms of the title, is within the scope of the journal.

Abstract

The abstract is brief but comprehensive, and it provides a structured overview including goal, results, conclusion, and implications of findings. However you wrote that (in line 16-18) "This study concludes that sufficient infrastructures and the location of resettlement areas affect residents’ sustainability perceptions (RSPs) of relocated communities, which in turn influence their SOC." But it is an assumption. In line 305 it is written that "analytical results for Rinari may be due to its sufficient infrastructures", or in line 313-314 you wrote that "These analytical results might be explained by the location of the resettlement site". Please consider it, and correct or reword it.

Introduction

This section introduces the study well, giving an account of the significance, topicality and purpose of the research.

Theoretical Framework

This section is very well-based that gives a perfect context for the justification of the research. This section includes many relevant references and the authors provide theoretical foundations for the analysis using appropriate references.

Five hypotheses were formulated in the research. Four of them are related to the four component of sustainability perceptions and their effect on the sense of community. The fifth hypothesis is about the effect of SOC on SSCD.

Methodology

This section is well-based, correct, traceable, and consistent.

Results

This section is also well-based, well presented and well explained.

Discussion and Conclusions

This sections relate to the analysis and are comprehensive, while contribute to the objectives set out in the introduction section.

The section makes clear the paper’s added value to the existing knowledge and point out the theoretical contributions, which is confirmed by the conclusion. Practical or as it is written in the text managerial implications are also provided.

Overall

It is a well-designed and well-established research. The study is well-structured, consistent, and it is a comprehensive work. The rationale behind the research is well-explained, and the significance of the research is also well-illustrated. It presents well the role and importance of the examined factors, the relationships and the context among them. The goals are clear and forward-looking. The methods and the analysis are excellent. The study reveals some important aspects of the topic.

I spotted three typos in the figures presented. In Figure 1, 2 and 3 “SSCD” hed been misspelt as “SCSD”.

Overall Recommendation

Overall, the manuscript can be accepted after minor revision.

Recommendation: minor revision

Author Response

Reviewer 2

Overall Comments:

The study is about to investigate the influence of sustainability perceptions on sense of community and support for sustainable community development in two Taiwanese relocated communities. The study proved that sustainability perceptions ifluencing the sense of community which can be considered as an antecedent of SSCD in both relocated communities. The authors found differences between the studied communities in terms of components that have a direct effects on the SOC. The topic of the paper is relevant and important. It is a fundamental issue for communities that are exposed to adverse weather conditions today and even more so in the future.

Responses from Authors:

Thank you very much for giving precise comments to improve my revised version of the manuscript (ijerph-1406919). We highlight the changes in the revised manuscript using green colored text. Our responses to your comments are as follows:

  1. Thank you for your supportive comment.

  1. Thank you for your helpful comment. We have changed the manuscript title to “Influence Analysis of Sustainability Perceptions on Sense of Community and Support for Sustainable Community Development in Relocated Communities”.

P.1; line 2-4

  1. Thank you for your helpful comment. We have revised the abstract of the manuscript as follow:
  2. 1; line 10-20

Abstract: This study aims to examine a theoretical model using sustainability perceptions including environmental, sociocultural, economic and life satisfaction; sense of community; and support for sustainable community development among the Indigenous people of two relocated communities in Taiwan. A total of 747 usable questionnaires were collected and analyzed using structural equation modeling. The analytical results indicated that sense of community is an antecedent of support for sustainable community development in both relocated communities. Life satisfaction perceptions can influence the sense of community in Rinari. Additionally, environmental and economic perceptions are antecedents of the sense of community in New Laiyi. Finally, this study provides theoretical implications to fill the gaps in previous research, and offers valuable insights for promoting residents’ support for sustainable community development in aboriginal communities, thus, this study has significant contributions theoretically and practically.

  1. Thank you for your encouragement and supportive comment.

  1. Thank you for your encouragement and supportive comment.

  1. Thank you for your encouragement and supportive comment.

  1. Thank you for your encouragement and supportive comment.

  1. Thank you for your encouragement and supportive comment.
  2. Thank you for your encouragement and supportive comment. In addition, we have revised all typos in main text.

Reviewer 3 Report

The paper presents a topical issue and emphasizes the importance of residents’ sustainability perceptions in enhancing their sense of community for promoting the support for sustainable community development in aboriginal communities. The paper contains interesting ideas and useful and relevant recommendations.

The structure of the article is clear. The concepts, the measurements and the methods are clearly defined and presented.

I have a few small remarks.

The research objective should be more clearly stated in the abstract, specifying what the theoretical model is for, and in accordance with the purpose suggested in the Introduction (Lines 50-53). At the same time, the objective of the study should be more clearly highlighted in the introductory section, too.

I recommend the authors to present in more detail the relationships identified in the literature between residents’ sustainability perceptions and their sense of community (Lines 120-122).

I also suggest the authors to strengthen the arguments for the formulated H1 – H4 hypotheses.

In the Discussion section, the authors should place more emphasis on discussing the different results obtained for the two analyzed sites.

Conclusions should include more convincing arguments regarding the contributions of the manuscript. I would also recommend adding a specific paragraph on the limitations of your study and future work.

I also noticed a repeated typo in the three figures (it is written SCSD instead of SSCD) that the authors may want to correct.

I hope you will find my comments useful in further improvement of your manuscript.

Author Response

Reviewer 3

Overall Comments:

The paper presents a topical issue and emphasizes the importance of residents’ sustainability perceptions in enhancing their sense of community for promoting the support for sustainable community development in aboriginal communities. The paper contains interesting ideas and useful and relevant recommendations.

The structure of the article is clear. The concepts, the measurements and the methods are clearly defined and presented.

Responses from Authors:

Thank you very much for giving precise comments to improve my revised version of the manuscript (ijerph-1406919). We highlight the changes in the revised manuscript using brown colored text. Our responses to your comments are as follows:

1.Thank you for your helpful comment. We have revised the abstract and introduction of the manuscript, and make the research objective more clearly.

  1. Thank you for your helpful comment. We have presented more detail and previous research about the relationships in the literature between residents’ sustainability perceptions and their sense of community. The revised manuscript sees as follow:
  2. 4; line 174-183

Moreover, some studies have identified causal relationships between different perceptions and attachment/sense of community. For example, Bonaiuto et al. [53] indicated that sociodemographic and residential variables might influence residents’ attachment, and length of residence in the place and economic perceptions are the most relevant residential and sociodemographic variables. Rollero and De Piccoli [19] studied 328 residents of a city in northern Italy, and the results indicated that individuals’ cultural level and social relationships were associated with place attachment. Mesch and Manor [16] argued that residents’ local attachment might result from a positive environmental perception. Stedman et al. [26] indicated that attachment to lakes is based in part on perceived quality of life and perceptions of social conflict.

  1. Thank you for your helpful comment. We have strengthened the arguments for the formulated H1 – H4 hypotheses. The revised manuscript sees as follow:
  2. 4-5; line 195-198

Moreover, some studies have indicated that different perception variables (e.g., economic, sociocultural, environmental, and life satisfaction perceptions) might influence place attachment or sense of place [16, 19, 26, 53].

Moreover, some studies have indicated that different perception variables (e.g., economic, sociocultural, environmental, and life satisfaction perceptions) might influence place attachment or sense of place [16, 19, 26, 53].

  1. Thank you for your helpful comment. We have described more details about the differences between two communities in Discussion and Conclusions section as follow:
  2. 12-14; line 472-483,502-508

In New Laiyi, economic perceptions and environmental perceptions were antecedents of sense of community, which is in agreement with the results of several previous studies [53, 81]. However, sociocultural perceptions and life satisfaction perceptions did not directly influence residents’ sense of community in New Laiyi. These analytical results might be explained by the location of the resettlement site. New Laiyi is located near Chaozhou township, the second-largest township in Pingtung County, with more than 53 thousand residents [83]. A large city can provide more working opportunities, which in turn improve the financial status and spending power of families. Several facilities that were considered iconic features in the Indigenous community, such as farms, flagstone houses, thatched granaries, and taro kilns, were reconstructed in New Laiyi [84]. These facilities may have not only maintained the community’s traditional customs but also improved residents’ connection with the place [85, 86].

According to attachment theory, different life experiences might influence individuals’ sense of community and influence their behavioral intention [24]. In this study, we found that different locations, psychological and physical benefits, and established schools in the two Indigenous communities might influence residents’ daily life experiences and perceptions, leading to different effects on the sense of community in the two different study sites. This study thus fills research gaps by extending the application of attachment theory in the community research context.

  1. Thank you for your helpful comment. We have replenished the 5.3. Limitations and future research directions section on the limitations of research and future work. In addition, we also revised 5.4. Conclusions section to convince more arguments regarding the contributions of the manuscript. The revised manuscript sees as follow:

  1. 13-14; line 545-569, 582-591

5.3. Limitations and Future Research Directions

Although our findings potentially make several contributions, there are several limitations that should be addressed in future research. First, this study applied four variables to measure residents’ sustainability perceptions. However, several studies have proposed other components of individuals’ sustainability perceptions. For example, Gattig and Hendrickx [95] argued that residents’ risk perceptions, including environmental, financial, and health risks, are an important component of sustainability perceptions. Holladay and Powell [96] proposed social-ecological resilience, including trust, networks, local control, flexible governance, leakage prevention, and controlled infrastructure development, to assess sustainability perceptions. Further research using these variables to more precisely assess residents’ sustainability perceptions is recommended.

Second, although this study provides several theoretical and managerial implications for two relocated communities, these suggestions might not be applicable to other relocated communities in Taiwan and other countries. To overcome this limitation, future research should assess the same theoretical model in relocated communities in different countries to verify its generalizability and to provide multicultural and/or international perspectives [97].

Finally, this study applied a self-reported survey to test a theoretical model. However, individuals’ perceptions might be influenced by their personal experiences [28], and some psychological or emotional attitudes toward community development may not actually be reflected in quantitative research methods [18]. To overcome this shortcoming, we recommend that future research use qualitative methods, such as in-depth interviews, focus groups, and participatory observations, to better capture the details of sustainable community development and the personal experiences that influence residents’ sustainability perceptions [18].

Therefore, this study suggests that administrators and developers should consider the different demands of each relocated community to improve the positive sustainability perceptions of residents, which in turn will increase their sense of community.

This study potentially makes contributions with both theoretical and practical implications and expands our knowledge in the research context of sustainability perceptions and the applicability of attachment theory in the community research context. The original theoretical framework proposed in this study sheds light on behavioral models for promoting support for sustainable community development in relocated Indigenous communities.

  1. Thank you for your helpful comment. We have revised all typos.

  1. Thank you for your helpful comment and encouragement.

Reviewer 4 Report

The manuscript "Influence Analysis of Sustainability Perceptions on Sense of Community and Support for Sustainable Development" addresses an important issue of significant concern to many: that is, what factors are influencing sustainability perceptions. The authors consider appropriate factors that might influence sustainability perceptions and use appropriate theoretical modeling and structural equation modeling to explore their take on what is influencing sustainability perceptions. The analysis is complex and the factors are complex and many of those who might use these results might find the theoretical and structural analysis challenging to understand. That said, the authors have put together and tested important ideas in a way that might move the field forward and that might be translatable in a form that would be understandable to many who would use this information but haven't had the opportunity to learn structural equation modeling.

Author Response

Reviewer 4

Overall Comments:

The manuscript "Influence Analysis of Sustainability Perceptions on Sense of Community and Support for Sustainable Development" addresses an important issue of significant concern to many: that is, what factors are influencing sustainability perceptions. The authors consider appropriate factors that might influence sustainability perceptions and use appropriate theoretical modeling and structural equation modeling to explore their take on what is influencing sustainability perceptions. The analysis is complex and the factors are complex and many of those who might use these results might find the theoretical and structural analysis challenging to understand. That said, the authors have put together and tested important ideas in a way that might move the field forward and that might be translatable in a form that would be understandable to many who would use this information but haven't had the opportunity to learn structural equation modeling.

Responses from Authors:

Thank you very much for giving precise comments to improve my revised version of the manuscript (ijerph-1406919). According to your suggestions, we have revised the main text to enhance the fluency and readability then ensure the reader understand the ideas of this research. Moreover, we have revised Discussion and Conclusions section to convince more arguments regarding the contributions of the manuscript.

Round 2

Reviewer 1 Report

Thank you for addressing the feedback